# Black Soldier Fly Larvae Meal as Alternative to Fish Meal for Aquaculture Feed

Marianna Oteri [1], Ambra Rita Di Rosa [1] , Vittorio Lo Presti [1], Filippo Giarratana [1] , Giovanni Toscano [2] and Biagina Chiofalo [1,*]

1   Department of Veterinary Sciences, University of Messina, 98168 Messina, Italy;
    marianna.oteri@unime.it (M.O.); dirosaa@unime.it (A.R.D.R.); vittorio.lopresti@unime.it (V.L.P.);
    filippo.giarratana@unime.it (F.G.)
2   Department of Chemical, Biological, Pharmaceutical and Environmental Sciences, University of Messina,
    98166 Messina, Italy; giovanni.toscano@unime.it
*   Correspondence: biagina.chiofalo@unime.it; Tel.: +39-0906766833

**Abstract:** *Hermetia illucens* meal (HIM) as ingredient in feed represents a way to achieve more sustainable food production. The aim was to characterize the chemical, microbiological and organoleptic characteristics of four diets for *Sparus aurata*, isoenergetic and isoproteic, containing 0%, 25%, 35% and 50% of HIM in substitution of fish meal (FM). Analyses were carried out using gas chromatography for fatty acids and amino acids, ICP-OES for minerals and liquid chromatography for aflatoxins and following International Organization for Standardization methods for microbial flora. E-sensing analysis of the diets was evaluated using an artificial sensory platform (E-eye, E-nose and E-tongue). The chemical results were submitted to a one-way ANOVA while Principal Component Analysis (PCA) of the e-sensing data was performed. No significant differences were observed for polyunsaturated fatty acids, thrombogenic and peroxidation indices among the diets. The replacement of FM with HIM increased the content of lysine, methionine, isoleucine, leucine, threonine and valine, while phosphorus, calcium and sodium content decreased ($p < 0.01$) as the percentage of HIM increased. Lead was significantly below the maximum level set by the EU regulation. The diets showed good hygienic and sanitary quality. The artificial senses permitted distinguishing color, odor and taste among the diets. Data allow considering *Hermetia illucens* as alternative protein source in fish nutrition.

**Keywords:** *Hermetia illucens*; aquaculture feed; fatty acids; amino acids; minerals; microbiological quality; e-sensing profile





## 1. Introduction

Among the costs of the aquaculture farms, feeding represents the largest portion (about 60%) of the economic balance; therefore, the economic success of fish production sector is mainly linked to the use of low-cost nutritionally balanced diets [1]. Therefore, the goal of aquaculture nutritionists, as well as fish farmers, is to obtain a very good conversion ratio to cover the cost of feeding. Proteins with a well-balanced presence of essential and non-essential amino acids are the most important nutrients for the maintenance, growth and feed efficiency of fish. They are the nutrients with the highest cost, and, therefore, their inclusion in the fish feed plays an important role in the overall feed costs. Fish meal, characterized by a high protein content, an excellent amino acid profile, a low carbohydrate level and a high digestibility, is considered the most important feeding source in fish nutrition [2]. However, the significant increase in the price of fish meal, the high protein nutritional needs of fish and the alternating availability in fish meal supply, in recent decades, have led to the study of alternative protein sources in aquafeeds [3].

In relation to the increasing demand for protein sources in animal feeding, there is a great interest towards unconventional sources. In recent years, industrial by-products,

co-products, insects, seaweed ingredients [4] and ex-food or former food products have been investigated as alternative ingredients for livestock and aquaculture feeding.

Interest in insects as feed ingredient for terrestrial and aquatic animals continues to grow [5–8]. At present, the exploiting of insects as feed ingredients is not in direct competition with food production. The use of insect meal as a substitute of fish meal seems to represent interesting perspectives for limiting the environmental impact of some production system, such as aquaculture, and for contributing to a circular economy and a "zero waste" society [9].

Among the insect species used as unconventional protein source for fish feeding, *Hermetia illucens* L. is of the most interesting source for its sustainability related to its capacity to convert organic waste material into biomass containing proteins (40–45%) with high biological value [10], fat (30–35%) with fatty acids of nutritional interest and ash (11–15%) with high mineral concentrations and a high Ca/P ratio [11]. In terms of protein quality, *Hermetia illucens* larvae contain a favorable essential amino acid profile closer to fish meal than that of soybean meal [5]. The mineral profile and fatty acid composition of the *Hermetia illucens* larvae were found to be influenced by the diet [12–14]. Liland et al. [14] reported that *Hermetia illucens* larvae are unable to synthesize polyunsaturated fatty acids, and, therefore, the presence of linoleic acid and alpha-linolenic acids, as well as of eicosapentaenoic and docosahexaenoic in the larvae most likely originates from the substrate. Furthermore, these authors observed that the presence of seaweed in the feeding media can enrich the larvae with macro- and microelements, making the insects a good source of minerals [14].

Nevertheless, Weththasinghe et al. [15] observed a linear decrease in protein and lipid digestibility, protein efficiency ratio and lipid retention in extruded diets for Atlantic salmon as the level of dietary *Hermetia illucens* meal increases.

The use of *Hermetia illucens* larvae meal (HIM) as a component of feed is a way to achieve more sustainable food production. According to current regulation in Europe [16], the use of HIM is mainly permitted in aquaculture. The optimal level of dietary substitution of fish meal for HIM varies considerably across studies, ranging from 25% to 100%, probably in relation to the different quality of larvae meal, fish species and diet formulation. Furthermore, to obtain a higher-quality fish, it is important to provide good quality and pathogen-free feeds to the fish [17]. In this view, the extrusion method destroys undesirable microbial flora, enzymes and anti-nutritional factors [18–20] and improves the nutritional value of final products and the apparent absorption of minerals [21,22].

Fish have strongly developed chemosensory and chemical signaling systems due to their living in an aquatic environment. The olfactory and gustatory systems comprise the main chemosensory pathways [23]. The combination of artificial senses (Electronic Nose, an Electronic Tongue and an Electronic Eye) proves to be a powerful tool to distinguish different organoleptic profiles related to the different chemical compositions of aquaculture feeds [24].

To further characterize the feasibility of HIM as a unconventional protein source in aquafeeds, the aim of this study was to characterize the chemical, microbiological and mineral composition of diets for *Spaurus aurata* L. containing HIM as a partial replacement for fish meal (FM). The effects of inclusion of HIM on the organoleptic characteristics of the diets were also investigated with the aim of providing detailed sensory information on the HIM-containing fish meal useful for feed industry.

## 2. Materials and Methods

Four experimental diets were formulated to satisfy the nutritional needs of *Sparus aurata*. The diets were isoenergetic (about 22 MJ/kg gross energy), isonitrogenous (about 43 g/100 g, as fed) and isolipidic (about 19 g/100 g, as fed). A basal diet (HIM0) with fish meal (FM), as exclusive protein source of animal origin, was prepared. FM was partially replaced with defatted *Hermetia illucens* meal at 25%, 35% and 50% (as fed basis) in the other three diets (HIM25%, HIM35%, HIM50%) corresponding to the inclusion levels of

0%, 7.9%, 11% and 15.7% of HIM, respectively. The other components of the formulas were adapted to obtain diets with the same energetic content.

Diets were prepared by SPAROS Lda (Olhao, Portugal); all dietary ingredients were ground, mixed and extruded using die with 4 mm diameter; the oils were added using a vacuum coating technology. The ingredients of the diets and the proximate composition of the diets (HIM0, HIM25%, HIM35% and HIM50%) are reported in Table 1.

**Table 1.** Diet ingredients and proximate composition of the experimental diets.

|  | HIM0 | HIM25% | HIM35% | HIM50% |
|---|---|---|---|---|
| Ingredients, % as fed |  |  |  |  |
| Fish meal | 25.00 | 18.75 | 16.25 | 12.50 |
| *Hermetia illucens* meal | 0 | 7.90 | 11.00 | 15.70 |
| Soy protein concentrate | 5.00 | 5.00 | 5.00 | 5.00 |
| Wheat gluten | 5.00 | 5.00 | 5.00 | 5.00 |
| Corn gluten | 5.00 | 5.00 | 5.00 | 5.00 |
| Soybean meal 48 | 15.00 | 15.00 | 15.00 | 15.00 |
| Rapeseed meal | 5.00 | 5.00 | 5.00 | 5.00 |
| Wheat meal | 17.45 | 15.17 | 14.21 | 12.88 |
| Whole peas | 4.00 | 4.00 | 4.00 | 4.00 |
| Fish oil | 5.00 | 5.00 | 5.00 | 5.00 |
| Rapeseed oil | 10.00 | 9.80 | 9.80 | 9.80 |
| Vitamin and mineral premix | 1.00 | 1.00 | 1.00 | 1.00 |
| Vitamin C35 | 0.03 | 0.03 | 0.03 | 0.03 |
| Vitamin E50 | 0.02 | 0.02 | 0.02 | 0.02 |
| Antioxidant | 0.30 | 0.30 | 0.30 | 0.30 |
| Sodium propionate | 0.10 | 0.10 | 0.10 | 0.10 |
| MCP, monocalcium phosphate | 1.50 | 2.20 | 2.50 | 2.80 |
| L-Lysine | 0.30 | 0.35 | 0.37 | 0.40 |
| L-Tryptophan | - | 0.03 | 0.04 | 0.05 |
| DL-Methionine | 0.10 | 0.15 | 0.18 | 0.22 |
| L-Taurine | 0.20 | 0.20 | 0.20 | 0.20 |
| Chemical composition, % as fed |  |  |  |  |
| Dry matter | 92.33 | 92.78 | 92.90 | 92.64 |
| Crude protein | 42.7 | 42.7 | 42.7 | 42.7 |
| Crude fat | 18.6 | 18.6 | 18.6 | 18.7 |
| Crude fiber | 2.3 | 2.2 | 2.2 | 2.1 |
| Ash | 9.3 | 9.3 | 9.4 | 9.3 |
| NFE * | 19.43 | 19.98 | 20.00 | 19.84 |

HIM0, fish meal; HIM25%, HIM35% and HIM50%, *Hermetia illucens* meal at 25%, 35% and 50% substitution rate of fish meal, respectively. * Nitrogen-free extract, NFE (%) = 100 − (%Crude Protein + %Crude fat + %Crude fiber + %Ash).

### 2.1. Fatty Acid Analysis

Triplicate feed samples were analyzed for the fatty acid composition. Each sample (ca. 2.5 g) added with sodium sulfate (1 g) was pounded manually. The lipids were extracted for 6 h with petroleum ether by a Soxtec™ 8000 Extraction system (FOSS, Padua, Italy). The fatty acid methyl esters (FAMEs) were produced from aliquots of lipids. In detail, sulfuric acid–methanol (1:9, *v/v*) reagent (2 mL) was added to the extracted lipid samples, and they were then heated at 100 °C for 1 h [25]. FAMEs were analyzed by a Trace 1310 chromatograph (Thermo Fisher Scientific, Milan, Italy) equipped with a flame ionization detector (FID) and a fused silica capillary column (30 m × 0.25 mm I.D., 0.25 μm film thickness) (Omegawax 250; Supelco, Bellefonte, PA, USA) maintained at 100 °C for 5 min, from 100 to 240 °C at 4 °C/min and final isotherm of 240 °C (20 min). Injector and detector temperatures were 250 °C. Injection volume and split ratio were 0.5 μL and 1:50, respectively. The carrier gas was helium at a flow rate of 1 mL/min. Data acquisition was carried out by a Chromeleon Software (Thermo Fisher Scientific, Milan, Italy). The identification of individual compounds was carried out by comparing their retention times

with those of standards (mix 37 FAMEs, Supelco, Inc., Bellefonte, PA, USA). The results were expressed as g/100 g of the total fatty acids identified.

Nutritional indices were calculated from the identified fatty acids, as proposed by Ulbricht and Southgate [26], for atherogenic (AI) and thrombogenic (TI) indices, while the equation proposed by Santos-Silva et al. [27] was used for the calculation of the hypocholesterolaemic/hypercholesterolaemic ratio (H/H). Indices were determined according to the following formulas:

$$\text{IA} = [\text{C12:0} + (4 \times \text{C14:0}) + \text{C16:0}]/(\Sigma\text{n6-PUFA} + \Sigma\text{n3-PUFA} + \Sigma\text{MUFA}) \tag{1}$$

$$\text{IT} = (\text{C14:0} + \text{C16:0} + \text{C18:0})/[(0.5 \times \Sigma\text{MUFA}) + (0.5 \times \Sigma\text{n6-PUFA}) + (3 \times \Sigma\text{n3-PUFA}) + (\Sigma\text{n3-PUFA}/\Sigma\text{n6-PUFA})] \tag{2}$$

$$\text{H/H} = (\text{C18:1n9} + \text{C18:2n6} + \text{C20:4n6} + \text{C18:3n3} + \text{C20:5n3} + \text{C22:5n3} + \text{C22:6n3})/(\text{C14:0} + \text{C16:0})] \tag{3}$$

Furthermore, the peroxidation index (PI), which expresses a measure of the peroxidation susceptibility and peroxidative lipid damage for a particular phospholipid membrane, was calculated using the formula reported below [28]:

$$\text{PI} = (\% \text{ dienoic} \times 1) + (\% \text{ trienoic} \times 2) + (\% \text{ tetraenoic} \times 3) + (\% \text{ pentaenoic} \times 4) + (\% \text{ hexaenoic} \times 5) \tag{4}$$

### 2.2. Amino Acid Analysis

Triplicate feed samples were analyzed for the amino acid composition. For the amino acid analysis, protein hydrolysis and derivatization were performed prior to the separation by GC-FID. Each sample (about 0.25 g) was hydrolyzed in 10 mL of a HCl solution (6 M) at 110 °C for 24 h. During the acid hydrolysis, the asparagine and glutamine were converted to aspartic and glutamic acids [29]; therefore, they were calculated as the sum of the aspartic acid plus asparagine and of the glutamic acid plus glutamine. For cysteine analysis, prior to the acid hydrolysis, a preliminary oxidation was performed for the deamination. Each sample was treated with formic acid:hydrogen peroxide (1:20, *v/v*) reagent (2 mL) for 30 min at room temperature. Then, a hydrolysis step with a HCl solution (6 M) was performed [30,31]. For the tryptophan analysis, each sample was hydrolyzed in 10 mL of a NaOH solution (4 M) at 112 °C for 16 h, and, after the hydrolysis, each sample was cooled and neutralized with acetic acid [32]. For the chromatographic analysis, procedures for purification, pre-column derivatization and qualitative and quantitative analyses of each amino acid were performed using the EZ:Faast Kit (Phenomenex, Torrance, CA, USA). A Trace 1310 chromatograph (Thermofisher, Waltham, MA, USA) was used, with a flame ionization detector (FID) and a ZB-AAA Amino Acid column (10 m × 0.25 mm ID); the oven temperature was programmed from 110 to 320 °C at 32 °C/min, with a final isotherm of 320 °C (1 min). Injector and detector temperatures were 250 and 320 °C, respectively. Injection volume and split ratio were 2.5 μL and 1:15, respectively.

### 2.3. Aflatoxin Analysis

The analysis of aflatoxins was carried out following the EN ISO method [33] and the criteria suggested by European regulation [34]. For the extraction of the aflatoxins, each sample (about 25 g) of the experimental diet was treated with 125 mL of a methanol:water mixture (7:3, *v/v*) and 5 g of NaCl. The extract was filtered, diluted with water and passed through an immunoaffinity column (Vicam) containing specific antibodies for aflatoxins B1, B2, G1 and G2. The aflatoxins were isolated, purified and concentrated on column and then recovered with methanol. The above procedure was performed in triplicate on each experimental diet. Aflatoxins were measured by a RP-HPLC coupled to a fluorescence detector (RF), using a post-column derivatization (Kobra cell system). Chromatographic separation was performed using a Nexera LC System (Shimadzu, Milan, Italy), equipped with a Luna column C18 250 × 4.6 mm (lenght × i.d.) and 5 μm of particle size (Phenomenex, Torrance, CA, USA). The mobile phase for the isocratic separation was

a mixture of water:acetonitrile:methanol (3:1:1, *v/v/v*) with 0.35 mL of nitric acid 4 M and 120 mg/L of potassium bromide. The RF wavelength was set at 365 nm for excitation and 435 nm for emission. The injection volume was 50 µL and the amount of aflatoxins was calculated using an external standard method. The LC method for aflatoxin analyses has been validated for the simultaneous chromatographic determination of total (B1, B2, G1 and G2) and B1 aflatoxins.

A calibration curve was constructed for each aflatoxin; the linearity was also tested in the range of 0.05–22 µg/kg, providing a correlation coefficient (R2) of $\geq$0.9996. The Limit Of Detection (LOD, between 0 and 0.05 µg/kg) and the Limit Of Quantification (LOQ, between 0.05 and 0.13 µg/kg) were calculated by the signal-to-noise (S/N) ratio, which should be greater than 3 and 10, respectively, according to the IUPAC criteria.

### 2.4. Mineral Element Analysis

Triplicate feed samples were analyzed for the mineral composition. Each sample was weighed (about 0.5 g) into an acid-prewashed PTFE vessels and 7 mL of HNO3 at 65% and a Rhenium internal standard (1 mL) were added. The mixture was digested with 1 mL of $H_2O_2$ at 30% using a closed-vessel microwave digestion system (Ethos 1, Milestone, Bergamo, Italy). To validate the analytical method, a Standard Reference Material of spinach leaves (SRM, NIST-1570a) obtained from the National Institute of Standards and Technology (Gaithersburg, MD, USA), was digested using the same analytical method described for the feed samples. An Avio200 ICP-OES instrument (Perkin Elmer, Waltham, MA, USA) equipped with a vertical DualView optical system and a S10 autosampler (Perkin Elmer, Waltham, MA, USA) was used to analyze the mineral content. Table 2 shows the recommended analytical lines length used to perform element analyses; the Argon line at 420.069 nm was used as an internal standard. The applied operational conditions are listed in Table 3. Data were processed using a PerkinElmer Syngistix™ for ICP software (Perkin Elmer, Waltham, MA, USA).

**Table 2.** Analytical lines length (nm) utilized for analysis.

| Element | nm | Element | nm | Element | nm |
|---------|---------|---------|---------|---------|---------|
| Cr | 267.716 | K | 766.490 | Se | 196.026 |
| Cu | 327.393 | Mg | 285.592 | Zn | 213.857 |
| Fe | 238.204 | Mn | 257.610 | Na | 589.592 |
| B | 249.677 | Ca | 317.933 | Pb | 220.353 |

**Table 3.** Operational conditions of the ICP-OES.

| Parameter | Conditions |
|-----------|------------|
| Radiofrequency power (W) | 1500 |
| Plasma gas flow (L/min) | 9 |
| Auxiliary gas (L/min) | 0.2 |
| Nebulizer gas (L/min) | 0.7 |
| Sample uptake (mL/min) | 1 |

The position of the torch was optimized prior to the analytical phase using the optical optimization procedure of Syngistix™ ICP software with Mn analytical line. All the quantitative measurements were made against external calibration curves constructed from a standard solution of 0.05, 0.25 and 1 ppm of Perkin Elmer (Waltham, MA, USA) for ICP analysis. A Milli-Q ultrapure (Merck Millipore, Merck KGaA, Darmstadt, Germany) water system was used to produce water at 1.8 MΩ/cm for the preparation of solutions and to dilute samples as needed. The calibration curves for all elements were established using the calibration blank and the reagent blank, and all of them resulted with correlation coefficients ($r^2$) better than 0.999; the Detection Limits (DLs) of this procedure were determined by

analyzing a matrix blank, which consisted of the same reagents and quantities as those used for sample preparation.

### 2.5. Microbiological Analysis

Twenty-five grams of each experimental diet were homogenized with buffered peptone water (Biolife, Milano, Italy) (ratio of 1:9 *w/v*) by using a stomacher (400 Circulator; International PBI s.p.a., Milano, Italy) for 60 s at 230 rpm. For each sample, the following parameters were evaluated: (i) enumeration of the aerobic colony at 30 °C [35] on Tryptic Glucose Yeast Agar (Biolife, Milano, Italy) plates, incubated at 30 ± 1 °C for 72 h; (ii) *Enterobacteriaceae* detection [36] and count [37] on Violet Red Bile Glucose Agar (Biolife, Milano, Italy), incubated at 37 ± 1 °C for 24 h; (iii) enumeration of coliforms [38] on Violet Red Bile Agar (Biolife, Milano, Italy) plates, incubated at 30 ± 1 °C for 24 h; (iv) yeasts and moulds count [39] on Dichloran Glycerol Agar (DG18 Biolife, Milano, Italy), incubated at 25 ± 1 °C for 5 days; (v) detection and enumeration of *Clostridium* spp. [40] on Tryptose Sulfite Cycloserine Agar (Biolife, Milano, Italy), incubated at 37 ± 1 °C for 24 h in anaerobic conditions; and (vi) detection of *Salmonella* spp. [41] on Chromogenic Salmonella Agar (Biolife, Milano, Italy) and Xylose Lysine Deoxycholate Agar (Biolife, Milano, Italy) incubated both at 37 ± 1 °C for 24 h. The limit of detection (LOD) was 10 CFU/g for the count of aerobic colonies at 30 °C, *Enterobacteriaceae*, coliforms, *Clostridium* spp. and 100 CFU/g for the count of yeasts and molds. Further 25 g of each experimental diet, as previously reported, were homogenized with Listeria Fraser Broth Half Concentration (Biolife, Milano, Italy) for the detection of the *Listeria monocytogenes* [42], incubated at 30 ± 1 °C for 20 h, followed by a passage in Listeria Fraser Broth (Biolife, Milano, Italy) at 37 ± 1 °C for 24 h and spread both on Agar Listeria according to Ottaviani & Agosti (ALOA®) (Biolife, Milano, Italy) and Listeria Palcam Agar (Biolife, Milano, Italy) both incubated at 37 ± 1 °C for 24–48 h.

### 2.6. E-Sensing Analysis

The feed samples were analyzed using an artificial sensory platform consisting of an E-eye, E-nose and E-tongue.

E-eye: The image was acquired with an artificial vision system (Iris visual analyzer 400, Alpha MOS, Toulouse, France) equipped with a high-resolution charge-coupled device (CCD) camera with 16 million colors. Image acquisition was performed using a top illumination and a white tray at the bottom to easily remove the background contribution (threshold selection applied: R 0-145, G 0-121, B 0-109). Each sample was ground, and the powder was placed and flattened on a plastic Petri dish (diameter 92 mm, height 7 mm). For each sample, 15 images on 15 freshly prepared dishes were acquired. Color spectra were calculated by selecting only the contributions greater than 1%.

E-nose: Odor analysis was performed by an electronic nose device (FOX 4000, Alpha MOS, Toulouse, France) equipped with 18 MOS (metal-oxide semiconductor) gas sensors and an automatic headspace sampler (HS100). For each sample, 15 replicates were prepared by placing 2 g of freshly ground fish feed into 10 mL headspace sealed vials. All parameters of the instrument are reported in Table 4.

**Table 4.** E-nose parameters.

| Acquisition | Oven |
|---|---|
| Duration 120 s | Incubation time 600 s |
| Period 1 s | Incubation temperature 40 °C |
| Time 1080 s | Syringe |
| Flow of the carrier gas 150 mL/min | Flushing time 120 s |
| Agitator | Temperature 50 °C |
| Speed 500 rpm | Fill speed 500 μL/s |
| On 5 s | Injection |
| Of 2 | Volume 500 μL |
| | Speed 500 μL/s |

E-tongue: Artificial taste analysis was performed using a commercially available electronic tongue (Astree, Alpha MOS, Toulouse, France) equipped with a set of seven potentiometric sensors (ANS, PKS, CTS, NMS, CPS, ANS and SCS), an Ag/AgCl reference electrode (Metrohm, Pte Ltd., Singapore), a mechanical stirrer and a 48-position autosampler. Five grams of each sample were ground and placed in 50 mL of deionized water for 15 min and centrifuged at 3000 rpm for 30 min. The solution was filtered and placed in a 25 mL beaker for the analysis. Single sample analysis was repeated 30 times to obtain the most stable sensor response and the last 15 measurements were considered to perform data processing. The signal was acquired every second for 120 s and the average intensity of the last 20 s was measured. Prior to measurement, the sensors were conditioned using one of the samples as a standard.

*2.7. Statistical Analysis*

The chemical data were analyzed by a one-way ANOVA, using the XLSTAT statistical package [43]. The percentage integration of insect meal (HIM0, HIM25%, HIM35% and HIM50%) was used as a fixed effect. Separation of means was assessed by Tukey's test, and differences were significant if $p < 0.05$.

A Principal Component Analysis (PCA) of the sensory profile data was performed by Alpha Soft V12.4 (Alpha-MOS, Toulouse, France) to evaluate the discrimination ability between the four experimental diets. The effectiveness of discrimination was assessed by evaluating the discrimination index (DI), which gives the quality of discrimination through an indication of the surface between groups. The DI is calculated automatically by the instrument's software according to the following formula:

$$DI = 100 \times [1 - [(Surface\ (A) + Surface\ (B) + Surface\ (C))/(Total\ Surface)]] \quad (5)$$

The DI reaches a maximum value of 100 when the groups are completely resolved.

**3. Results**

*3.1. Fatty Acid Profile*

The fatty acid profile of the four experimental diets is shown in Table 5. Saturated fatty acids did not show any significant ($p > 0.05$) differences, with the exception of the lauric acid (C12:0), which showed significantly ($p < 0.01$) higher values in the HIM35% and HIM50% diets than those observed in the HIM0 and HIM25% diets and for the palmitic acid (C16:0) which showed a significant ($p < 0.05$) higher value in the HIM0 diet than those observed in the diets containing *Hermetia illucens* meal. Monounsaturated fatty acids show similar content among the experimental diets as well as the polyunsaturated fatty acids, of both the omega 3 and omega 6 series. Table 6 shows the fatty acid classes; the sum of saturated, monounsaturated and polyunsaturated fatty acids; and some indices of nutritional interest, namely the atherogenic (AI) and thrombogenic (TI) indices, the peroxidation index (PI) and the hypocholesterolaemic/hypercholesterolaemic ratio. No significant ($p > 0.05$) difference was observed among the fatty acid classes, with the exception of the saturated fatty acids which show the highest ($p < 0.05$) level in the control diet (HIM0). The sum of

the eicosapentaenoic (EPA) and docosahexaenoic (DHA) acids was similar among the diets. Similar values were recorded for AI, TI and PI. The H/H ratio showed significantly ($p < 0.05$) higher values in the HIM35% and HI50% diets than that recorded in the HIM0 diet while the HIM25% diet showed a value of the H/H ratio similar to those of the other diets.

**Table 5.** Fatty acid composition (g/100 g of fatty acid methyl esters) # of the experimental diets.

| Fatty Acid | HIM0 | HIM25% | HIM35% | HIM50% | SEM | *p* |
|---|---|---|---|---|---|---|
| C10:0 | 0.04 | 0.03 | 0.05 | 0.05 | 0.008 | 0.391 |
| C12:0 | 1.12 B | 0.52 C | 1.67 A | 1.34 AB | 0.080 | 0.002 |
| C13:0 | 0.02 | 0.02 | 0.01 | 0.01 | 0.004 | 0.615 |
| C14:0 | 2.67 | 2.57 | 2.55 | 2.54 | 0.065 | 0.552 |
| C15:0 | 0.23 | 0.22 | 0.20 | 0.21 | 0.009 | 0.319 |
| C16:0 | 12.04 a | 11.43 ab | 11.19 b | 11.17 b | 0.106 | 0.013 |
| C16:1 | 3.10 | 3.01 | 2.87 | 2.99 | 0.121 | 0.629 |
| C17:0 | 0.20 | 0.20 | 0.19 | 0.19 | 0.008 | 0.615 |
| C18:0 | 2.72 | 2.79 | 2.64 | 2.54 | 0.100 | 0.445 |
| C18:1n9 | 43.49 | 44.69 | 43.26 | 43.55 | 0.328 | 0.116 |
| C18:1n7 | 3.21 | 3.19 | 3.10 | 3.11 | 0.041 | 0.263 |
| C18:2 n6 | 14.34 | 14.28 | 14.92 | 14.79 | 0.196 | 0.184 |
| C18:3n6 | 0.11 | 0.11 | 0.10 | 0.10 | 0.004 | 0.410 |
| C18:3n3 | 4.25 | 4.36 | 4.45 | 4.50 | 0.088 | 0.318 |
| C20:0 | 0.47 | 0.44 | 0.43 | 0.42 | 0.022 | 0.465 |
| C20:1n9 | 2.12 | 2.25 | 1.88 | 2.05 | 0.150 | 0.469 |
| C20:2n6 | 0.11 | 0.11 | 0.10 | 0.10 | 0.004 | 0.615 |
| C20:3n3 | 0.30 | 0.29 | 0.30 | 0.30 | 0.012 | 0.943 |
| C20:4n6 | 0.04 | 0.06 | 0.05 | 0.05 | 0.004 | 0.138 |
| C20:5n3 | 4.61 | 4.56 | 4.92 | 4.92 | 0.246 | 0.635 |
| C22:0 | 0.25 | 0.23 | 0.27 | 0.23 | 0.017 | 0.398 |
| C22:1n9 | 0.34 | 0.33 | 0.30 | 0.27 | 0.037 | 0.609 |
| C22:2n6 | 0.02 | 0.02 | 0.03 | 0.02 | 0.013 | 0.856 |
| C23:0 | 0.22 | 0.21 | 0.25 | 0.24 | 0.015 | 0.384 |
| C24:0 | 0.59 | 0.60 | 0.62 | 0.61 | 0.024 | 0.887 |
| C22:6n3 | 3.46 | 3.54 | 3.69 | 3.73 | 0.228 | 0.828 |

HIM0, fish meal; HIM25%, HIM35% and HIM50%, *Hermetia illucens* meal at 25%, 35% and 50% substitution rate of fish meal, respectively. # The concentration of fatty acid is expressed as g/100 g, considering 100 g the sum of the areas of all FAME identified. Mean values with different letters within the same row are significantly different, A–C at $p < 0.01$ and a and b at $p < 0.05$.

**Table 6.** Fatty acid classes, nutritional indices and in the experimental diets.

|  | HIM0 | HIM25% | HIM35% | HIM50% | SEM | *p* |
|---|---|---|---|---|---|---|
| SFA | 20.54 a | 19.23 b | 20.06 ab | 19.53 ab | 0.192 | 0.030 |
| MUFA | 52.25 | 53.46 | 51.40 | 51.97 | 0.549 | 0.198 |
| PUFA | 27.21 | 27.32 | 28.54 | 28.51 | 0.667 | 0.420 |
| n3-PUFA | 12.61 | 12.75 | 13.35 | 13.45 | 0.562 | 0.671 |
| n6-PUFA | 14.60 | 14.57 | 15.19 | 15.06 | 0.194 | 0.185 |
| EPA + DHA | 8.07 | 8.10 | 8.61 | 8.65 | 0.470 | 0.736 |
| AI | 0.30 a | 0.28 b | 0.29 ab | 0.29 ab | 0.004 | 0.032 |
| TI | 0.24 | 0.23 | 0.22 | 0.22 | 0.007 | 0.232 |
| PI | 59.50 | 60.00 | 62.90 | 63.10 | 2.376 | 0.633 |
| H/H | 4.77 b | 5.10 ab | 5.19 a | 5.21 a | 0.071 | 0.035 |

HIM0, fish meal; HIM25%, HIM35% and HIM50%, *Hermetia illucens* meal at 25%, 35% and 50% substitution rate of fish meal, respectively; SFA, saturated fatty acid class; MUFA, monounsaturated fatty acid class; PUFA, polyunsaturated fatty acid class; EPA, eicosapentaenoic acid; DHA, docosahexaenoic acid; AI, atherogenic Index; TI, thrombogenic Index; PI, peroxidation Index; H/H, hypocholesterolaemic/hypercholesterolaemic ratio. Mean values with different letters a and b within the same row are significantly different at $p < 0.05$.

### 3.2. Amino Acid Profile

Table 7 reports the amino acid composition of the experimental diets. Twenty amino acids were identified and quantified; ten of these belong to the indispensable amino acids and ten to dispensable ones. Among the indispensable amino acids, six amino acids (isoleucine, leucine, lysine, methionine, threonine and valine) showed significantly ($p < 0.01$) higher values in the diets containing insect meal, while histidine showed significantly ($p < 0.01$) higher values in the HIM35% and HIM50% diets than those observed in the HIM0 and HIM25% diets. Arginine, phenylalanine and tryptophan showed the highest ($p < 0.01$) values in the control diet (HIM0). Among the dispensable amino acids, hydroxylisine, hydroxyproline and tyrosine showed similar values among the experimental diets. Glutamic acid plus glutamine showed significantly ($p < 0.01$) higher values in all diets containing the *Hermetia illucens* meal, while proline and serine showed significantly ($p < 0.01$) higher values in the diets in which FM has been replaced with HIM at 35% and 50%. The levels of aspartic acid plus asparagine and that of glycine were significantly ($p < 0.01$) higher in the HIM25% and HIM35% diets than those observed in the HIM0 and HIM50% diets. Cysteine showed the highest ($p < 0.05$) value in the HIM25% diet and alanine the highest ($p < 0.01$) level in the HIM0 diet.

**Table 7.** Amino acid composition (g/100 g dry matter) of the experimental diets.

| | HIM0 | HIM25% | HIM35% | HIM50% | SEM | $p$ |
|---|---|---|---|---|---|---|
| Indispensable amino acids | | | | | | |
| Arginine | 2.84 A | 2.33 aB | 2.15 bB | 2.21 abB | 0.026 | <0.001 |
| Histidine | 1.10 B | 1.40 C | 1.60 A | 1.65 A | 0.023 | <0.001 |
| Isoleucine | 1.96 B | 2.42 A | 2.52 A | 2.38 A | 0.028 | 0.001 |
| Leucine | 3.40 bB | 4.13 aA | 4.22 aA | 4.04 bA | 0.024 | <0.0001 |
| Lysine | 4.66 B | 5.51 A | 5.59 A | 5.53 A | 0.084 | 0.004 |
| Methionine | 0.80 B | 0.94 A | 0.94 A | 0.94 A | 0.013 | 0.003 |
| Phenylalanine | 3.06 A | 2.31 C | 2.84 B | 2.87 B | 0.005 | <0.0001 |
| Threonine | 1.66 B | 2.09 A | 2.13 A | 1.96 A | 0.038 | 0.003 |
| Valine | 1.79 B | 2.35 A | 2.40 A | 2.26 A | 0.025 | <0.001 |
| Tryptophan | 0.12 A | 0.05 B | 0.05 B | 0.05 B | 0.002 | <0.0001 |
| Dispensable amino acids | | | | | | |
| Hydroxylysine | 0.21 | 0.24 | 0.26 | 0.23 | 0.013 | 0.149 |
| Alanine | 1.48 C | 2.04 A | 2.02 A | 1.81 B | 0.013 | <0.0001 |
| Aspartic acid + Asparagine | 2.37 C | 3.30 A | 3.16 A | 2.85 B | 0.037 | <0.001 |
| Cysteine | 0.21 b | 0.39 a | 0.19 b | 0.17 b | 0.018 | 0.006 |
| Glycine | 1.76 C | 2.30 A | 2.26 A | 2.04 B | 0.023 | <0.001 |
| Glutamic acid + Glutamine | 1.38 B | 1.64 A | 1.72 A | 1.74 A | 0.023 | 0.001 |
| Proline | 1.88 C | 2.44 B | 2.76 A | 2.82 A | 0.029 | <0.0001 |
| Hydroxyproline | 0.37 | 0.66 | 0.65 | 0.60 | 0.055 | 0.059 |
| Tyrosine | 1.22 | 1.39 | 1.54 | 1.57 | 0.071 | 0.075 |
| Serine | 2.49 B | 2.65 B | 2.95 A | 2.82 A | 0.028 | 0.001 |

HIM0, fish meal; HIM25%, HIM35% and HIM50%, *Hermetia illucens* meal at 25%, 35% and 50% substitution rate of fish meal, respectively. Mean values with different letters within the same row are significantly different, A–C at $p < 0.01$ and a and b at $p < 0.05$.

### 3.3. Mineral Element Profile

In Table 8, the average values of minerals in the experimental diets are reported as: macroelements (phosphorus, calcium, potassium, magnesium and sodium), whose needs by the body are in large amounts, microelements (copper, zinc, manganese, iron, etc.), whose needs by the body are in small amounts [44] and toxic metal (lead). The amount of phosphorus, calcium and sodium decreased significantly ($p < 0.01$) by increasing the amount of HIM to replace fish meal. The presence of insect meal over 25% as a substitute for fish meal (HIM35% and HIM50%) resulted in significantly ($p < 0.01$) lower levels of potassium, although the HIM35% and HIM50% diets showed similar ($p > 0.05$) values between them. The Ca/P ratio in the four experimental diets was calculated due to the antagonist effect of these two macroelements [45]. Significantly ($p < 0.01$) higher levels for

the Ca/P ratio were observed in the HIM0 and HIM25% diets than those of the HIM25% and HIM50% diets. Magnesium was significantly ($p < 0.01$) higher in the HIM25% diet than in HIM0, HIM35% and HIM50% diets, while these diets did not show ($p > 0.05$) differences among them. Iron, copper and zinc showed significantly ($p < 0.01$) higher levels in the HIM35% diet than those of the HIM0, HIM25% and HIM50% diets, except the HIM25% diet, which showed a similar content to the HIM50% diet. A significantly ($p < 0.01$) higher level of manganese was observed in the HIM25% diet than those of the HIM0, HIM35% and HIM50% diets. The increase in the level of HIM to replace fish meal resulted in a significant ($p < 0.05$) increase in the amount of chromium. Boron was significantly ($p < 0.01$) higher in the HIM25% and HIM35% diets compared to the other diets; lead showed the highest ($p < 0.01$) value in the HIM50% diet (Table 8).

**Table 8.** Mineral element profile (mg/kg dry matter) of the experimental diets.

| Items | HIM0 | HIM25% | HIM35% | HIM50% | SEM | *p* |
|---|---|---|---|---|---|---|
| Macroelements | | | | | | |
| P—Phosphorus | 11,452.63 A | 10,966.03 B | 10,477.43 C | 9557.83 D | 10.30 | <0.0001 |
| Ca—Calcium | 19,560.64 A | 18,673.81 B | 16,710.51 C | 15,064.18 D | 172.16 | <0.0001 |
| K—Potassium | 11,632.62 A | 11,832.99 A | 10,553.62 B | 10,262.06 B | 136.03 | <0.0001 |
| Mg—Magnesium | 2045.39 B | 7600.68 A | 2041.30 B | 1990.07 B | 78.62 | <0.0001 |
| Na—Sodium | 8081.56 A | 7575.17 B | 5664.86 C | 4675.18 D | 90.82 | <0.0001 |
| Ca/P ratio | 1.71 A | 1.70 A | 1.60 B | 1.57 B | 0.017 | 0.001 |
| Microelements | | | | | | |
| Fe—Iron | 219.84 C | 232.19 B | 250.24 A | 229.66 BC | 2.38 | <0.001 |
| Zn—Zinc | 175.25 C | 184.76 B | 192.10 A | 184.47 B | 1.18 | <0.0001 |
| Mn—Manganese | 7.13 D | 10.48 A | 9.86 C | 10.14 D | 0.062 | <0.0001 |
| Cu—Copper | 14.08 C | 15.20 BC | 16.56 A | 16.03 AB | 0.27 | 0.001 |
| Se—Selenium | 3.90 B | 4.63 A | 3.26 B | 2.84 B | 0.35 | <0.001 |
| Cr—Chromium | 4.43 b | 5.44 ab | 5.62 ab | 6.03 a | 0.28 | 0.019 |
| B—Boron | 6.49 B | 7.82 A | 7.72 A | 6.60 B | 0.18 | 0.001 |
| Toxic metal | | | | | | |
| Pb—Lead | 0.76 B | 0.97 B | 0.78 B | 1.83 A | 0.44 | 0.001 |

HIM0, fish meal; HIM25%, HIM35% and HIM50%, *Hermetia illucens* meal at 25%, 35% and 50% substitution rate of fish meal, respectively. Mean values with different letters within the same row are significantly different, A–D at $p < 0.01$ and a and b at $p < 0.05$.

### 3.4. Mycotoxin Profile

The amounts of total aflatoxins (B1, B2, G1 and G2) and aflatoxin B1 in the four experimental diets (HIM0, HIM25%, HIM35%, HIM50%) were below the limit of detection (LOD: 0.05 µg/kg) in all experimental diets.

### 3.5. Microbiological Profile

No significant difference on the load of each microbiological parameter was observed in the experimental diets. For all diets, only a few aerobic colonies with a load below 70 CFU/g (HIM0 = 65 CFU/g, HIM25% = 70 CFU/g; HIM35% = 50 CFU/g; HIM50% = 60 CFU/g) was observed. The counts of *Enterobacteriaceae*, coliforms, *Clostridium* spp., yeasts and molds were always under the LOD. *Enterobacteriaceae*, *Salmonella* spp., *L. monocytogenes* and *Clostridium* spp. were not detected.

### 3.6. E-Sensing Profile

Concerning data provided by the artificial sensory platform, the first step was to separately perform PCAs on the data from E-nose and E-tongue sensors and on the colors of the E-eye code. The next step was to look for the most effective way to combine the data provided by the E-eye, E-nose and E-togue in order to improve the discrimination capability. An intermediate fusion level was adopted in this study. The sensor data with the highest discrimination power were chosen; in particular, the data of four E-nose sensors (LY2/G, LY2/AA, P30/1 and T40/1), three E-tongue sensors (AHS, CTS and NMS) and

four colors extracted from the E-eye (codes 1620, 1890, 1891 and 2147) were chosen. These datasets were reduced, due to the different data size, and a new PCA was performed and the Discrimination Index (DI) calculated. The result is shown in Figure 1.

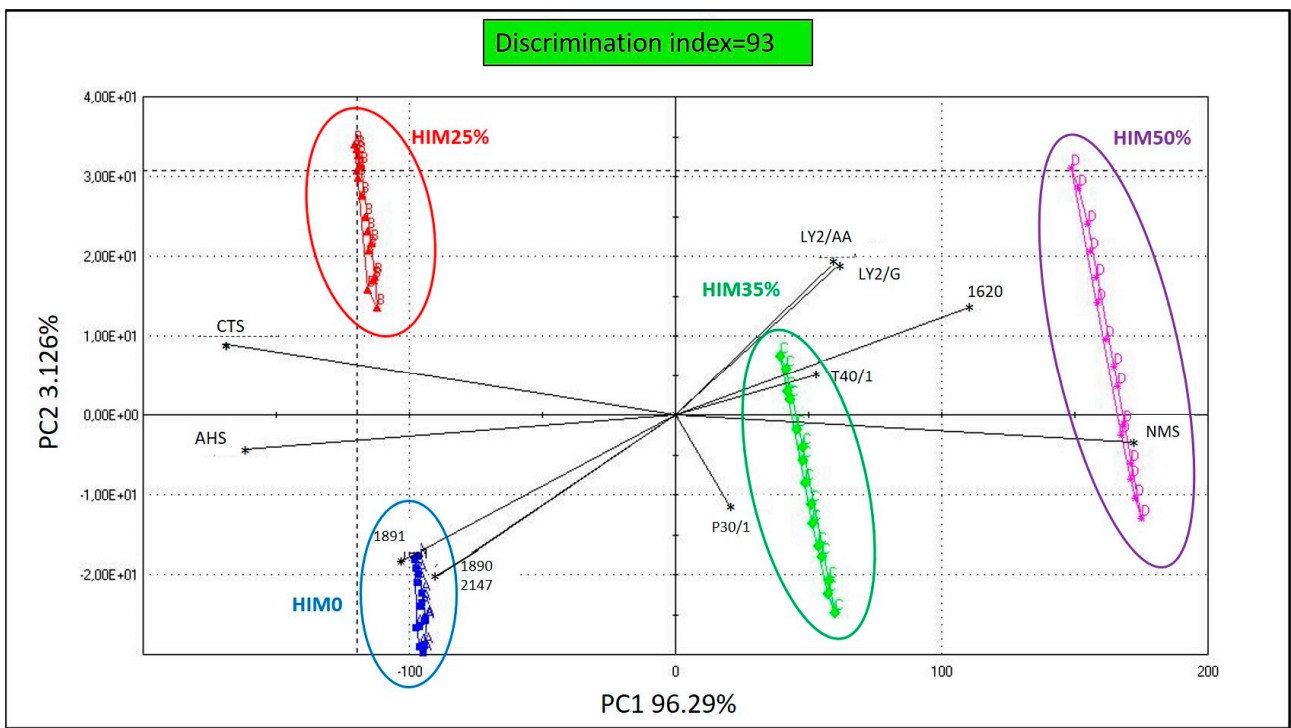

**Figure 1.** Principal component analysis map for fish feed groups (HIM0, blue; HIM25%, red; HIM35%, green; HIM50%, violet) and loading vectors of selected variables. At the top DI = 93.

The first two principal components calculated (PC1 and PC2) explain 99.4% of the total variance and show a DI of 93, highlighting a clear difference among the four diets, in relation to the substitution of fish meal with HIM. The combination of sensor responses with the highest discrimination power of the three artificial senses completely separates the four feed groups. Moreover, the loading plot helps identify the variables accountable for clustering of our dataset, by providing a numerical value which represents the contribution of each original variable to the score plot. Along the PC1 axis, from left to right, the groups are separated mainly by color codes 1890, 1891 and 2147, which are the most distinctive for the HIM0 group. On PC1, the volatile component of the different groups is also distinguished. Figure 1 shows the contribution of LY2/G, LY2/AA, P301 and T401. These sensors show a clear separation between HIM35% and HIM50% diets and the HIM0 and HIM25% diets. Regarding the taste profile, the three selected sensors (AHS, CTS and NMS) help to improve discrimination between groups. In particular, the AHS and CTS sensors have an important contribution on the HIM0 and HIM25% diets, while the NMS sensor mainly distinguishes the groups with higher insect meal content (HIM35% and HIM50%).

## 4. Discussion

Regarding the origin of fatty acids in the larvae, some studies report the possibility of modifying the fatty acid profile through the diet, while others the possibility of an endogenous synthesis of fatty acids. Knowledge of the fatty acid pathway is of great importance if HIM is used as an ingredient in animal feed. Our results are similar to the fatty acid profile determined by Belghit et al. [9] in four experimental diets formulated for Atlantic salmon with an increasing substitution of fish meal with HIM. St-Hilaire et al. [13] observed a reduction of alpha-linolenic acid (C18:3n3), eicosapentaenoic acid

(C20:5n3) and docosahexaenoic acid (C22:6n3) in fish fed a diet containing HIM. This could represent a problem for producers and consumers of the seafood supply chain [46]. However, our results show similar content of the alpha-linolenic acid, eicosapentaenoic acid and docosahexaenoic acid among the diets, despite the increasing addition of HIM in the feed. The content of these fatty acids, essential for the growth and development in fish [47] and associated with disease prevention and health promotion for humans, thanks to the production of anti-inflammatory eicosanoids [48], could be associated with the use, in feeding of the larvae, of a substrate containing fish offal and algae [12,14], since *Hermetia illucens* is not able to synthesize PUFA [14]. In fact, only plants and marine algae possess the enzymes necessary for the synthesis of linoleic acid (C18: 2n6) and alpha-linolenic acid (the precursor of EPA and DHA) [49]; therefore, similar concentrations of n6-PUFA (mainly linoleic acid) and n3-PUFA among the four diets confirm our hypothesis on the influence of the feeding substrate of *Hermetia illucens* larvae [46]. Another interesting result concerned the similar values recorded for the health lipid indices (atherogenic and thrombogenic indices), the peroxidation index and the H/H ratio. The atherogenic and thrombogenic indices (AI and TI) take into account, in the formulas used for their calculation, the contribution that each fatty acid has on human health and, in particular, on the probability of influencing the incidence of cardiovascular diseases [26]. The decrease in PI, e.g., during feed storage, indicates the oxidative degradation of PUFA in primary and secondary oxidation products. This process results in a loss of shelf-life, nutritional value and feed safety, as well as reduced consumer acceptability [50]. Finally, the low H/H ratio observed in diets containing HIM suggests some positive nutritional effects on animals.

As regards the indispensable amino acids, the substitution of fish meal with *Hermetia illucens* increased the lysine (EAA) and methionine content in all the levels of inclusion (HIM25%, HIM35% and HIM50%). In fact, *Hermetia illucens* meal has high levels of amino acids such as methionine and lysine [30]. Furthermore, isoleucine, leucine, threonine and valine showed higher levels in all experimental diets containing *Hermetia illucens* meal; this fact is nutritionally interesting, as fish cannot synthesize these amino acids de novo [51]. Among the dispensable amino acids (NEAA), the inclusion of HIM at 35% and 50% levels resulted in an increase in proline and tyrosine (NEAA), as observed by De Marco et al. [30] and by Iaconisi et al. [51].

Minerals are present in low quantities in the feed despite having important metabolic roles [22,44]. Mineral elements in fish are involved in many biochemical processes as enzyme cofactors and activators, in the formation of skeletal muscles, in the transmission of nerve impulses and in the acid–base chemical balance [44,52,53]. The low levels of calcium, potassium and sodium observed in the formulation containing *Hermetia illucens* meal do not represent a problem in the feeding of fish that can readily obtain these minerals from the surrounding water environment [44,53]. On the contrary, the phosphorus requirement must be satisfied through the diet as the fish cannot readily obtain it from the surrounding aquatic environment, which contains a low amount of it [54,55]. Phosphorus and calcium are generally combined together in the fish body, so an adequate Ca/P ratio in the diet is crucial to ensure healthy bone formation and growth performance of fish [45,52]. In relation to their antagonistic effect [56], the dietary Ca/P ratio has to be taken into consideration, as an excess of Ca or P can cause mineral imbalances that can affect the absorption of other minerals (e.g., an excess of Ca into the diet can reduce the absorption of Zn, Fe and Mn), causing some important consequences for bone development, which can be adversely affected when this ratio increases, causing anomalies in mineral homeostasis and bone mass [57] or environmental contamination (excess P is excreted). In our study, the Ca/P ratio of all diets was close to 1, as recommended for fish [58,59].

As for the microelements, such as zinc, essential for the growth and development in fish, cofactor of many enzymes and necessary for the activity of the antioxidant enzyme superoxide dismutase [44,60]; iron, necessary for the blood and muscle pigments [44] and actively involved in oxidation-reduction reactions [60]; and copper, involved in enzymatic activities and oxygen transportation [60], the highest levels were observed in the diet

containing a quote of 35% of defatted *Hermetia illucens* meal. This result would seem to highlight that the HIM35% diet is the best formula from a nutritional point of view, considering also that fish cannot obtain adequate quantities of these microelements from water [59,61] and that they must receive them through the diet [55].

The marine environment can convey toxic metals, naturally contained or introduced by various human activities through seafood [62]. Fish, both freshwater and marine water, that are at the top of the food chain, are extremely sensitive to exposure to lead (Pb), which is a highly toxic metal [63]. Its toxicity depends on various factors such diet and environment [63]. It seems that *Sparus aurata* is much more sensitive to waterborne lead exposure than in the diet [64]. However, considering the toxicity of lead, the European regulation [65] has set a maximum level of lead for complete animal feed of 5 ppm, which is considerably higher than the values found in our experimental diets.

Fish feeds become rapidly colonized by environmental microbes. Furthermore, all insects are colonized by microorganism and the insect microbiota is generally different from microorganism in the external environment, including ingested food [66,67]. Usually, the presence in commercial fish feeds of spoilage and pathogen bacteria such as *Salmonella* spp. and *Escherichia coli* can be related to poor hygienic storage condition, environmental contamination and problems in the extrusion treatment [68,69]. Our results on the microbiological profile of all diets confirm the effectiveness of the extrusion treatment in reducing the microbial load. The low charge observed for aerobic bacteria could be related to a secondary contamination after extrusion treatment. As recent studies have shown that aflatoxin contamination of animal feeds is a frequent issue, the European regulation has set maximum residue limits for total aflatoxins (B1, B2, G1 and G2) and aflatoxin B1 in animal feed [70]. To confirm the effectiveness of extrusion treatment and the good environmental condition during the storage, the total aflatoxins (B1, B2, G1 and G2) and aflatoxin B1 were found to be below the limit of detection in all experimental diets [71]. This is of particular interest because, in aquaculture, their presence in the diet can destroy the availability of certain nutrients, such as vitamin C and thiamine, reducing the immune defense of fish [72]. In addition, the contamination by mycotoxins of fish feed has been reported to cause intoxications [69], tissue abnormalities or liver injury, liver tumor, decreased growth rate and appetite [73] with a decrease of production efficiency and weight of the caught product and an increase in medical costs [74].

The E-nose and E-tongue sensors are non-specific and partially cross-sensitive; therefore, the value of each sensor is not directly related to each other. To better understand the data obtained by the artificial senses and improve the discrimination capability, it is necessary to optimize the analysis through a fusion process often categorized in a three-level model which includes the distinction of low, intermediate and high fusion levels [75,76]. In our study, the use of an intermediate fusion level helped improve the discrimination power. First, some relevant features were extracted from each data source separately, and then they were concatenated into a single array, which was used for multivariate classification and regression [77]. Selected color codes and E-nose and E-tongue sensors identified differences between groups in relation to the percentage of HIM integration. The volatile component was mainly distinguished by LY2/G, LY2/AA, P301 and T401, which are sensors primarily sensitive to volatile organic compounds derived from proteins and lipids such as nitrogen-containing compounds, hydrocarbons and aldehydes [78,79]. These sensors showed a clear separation of the HIM35% and HIM50% groups from HIM0 and HIM25%, in relation to the amino acidic content of the diets. Our observations appear interesting as the use of alternative ingredients in fish diets, in relation to their chemical composition and their level of inclusion, can reduce the acceptability of feeds, even if nutritionally balanced. In fact, different chemical substances can influence fish feeding behavior by acting as attractants through smell or taste [80]. Fish, in general, have a well-defined olfactory sensitivity to amino acids [81–85], which help them locate and identify food. Glycine and alanine are potent odorants that can stimulate feeding behavior by increasing food intake [86,87]. Our results show that HIM-integrated feeds had greater amounts of glycine and alanine com-



pared to HIM0 diet; this probably contributed to the olfactory discrimination performed by the E-nose sensors. Regarding the taste profile, the three sensors selected mainly represent umami taste (NMS), sourness (AHS) and saltiness (CTS). The results of PCA show that the HIM0 and HIM25% diets have a more salty and sour taste, probably related to their high sodium content, while the diets containing insect meal showed higher percentages of glutamic and aspartic acids, representative of the umami taste [88], than those of the HIM0 diet.

## 5. Conclusions

Data suggest that inclusion of *Hermetia illucens* meal positively influenced the hypocholesterolaemic/hypercholesterolaemic ratio and the content of indispensable amino acids and microelements. The microbiological quality of all diets testifies to the good practices of hygiene and sanitation applied during the production processes of fish feeds. The E-sensing analysis permitted distinguishing color, odor and taste in the four feed groups. The combination of sensor responses (E-eye, E-nose and E-tongue) proves to be a powerful tool for discriminating different organoleptic profiles linked to different chemical compositions of experimental diets.

This study represents a part of a larger investigation aimed at evaluating the suitability of HIM addition in the *Sparus aurata* diet through the study of the productive performance, the chemical and organoleptic characteristics of the fillets and the possible development of intestinal inflammation.

**Author Contributions:** Conceptualization, B.C.; methodology, A.R.D.R. and B.C.; software, A.R.D.R.; formal analysis, M.O., A.R.D.R., V.L.P., F.G. and G.T.; investigation, B.C.; data curation, A.R.D.R. and B.C.; writing—original draft preparation, M.O., V.L.P., F.G., A.R.D.R. and B.C.; writing—review and editing, V.L.P., A.R.D.R. and B.C.; supervision, B.C.; and funding acquisition, B.C. All authors have read and agreed to the published version of the manuscript.

**Funding:** This research was funded by PO FEAMP 2014–2020 mis. 2.47 CUP J46C18000570006, project codex 03/INA/17 Title of the project "FIFA—Feed Insects for Aquaculture", Scientific Responsible Biagina Chiofalo.

**Institutional Review Board Statement:** Not applicable.

**Informed Consent Statement:** Not applicable.

**Data Availability Statement:** Not applicable.

**Conflicts of Interest:** The authors declare no conflict of interest.

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
