# Peer review of "Black Soldier Fly Larvae Meal as Alternative to Fish Meal for Aquaculture Feed"

_sustainability, doi:10.3390/su13105447_

Round 1

Reviewer 1 Report

The manuscript entitled: ‘Black Soldier Fly Larvae meal as alternative to fish meal for aquaculture feed’ is well-written, interesting manuscript presenting original and useful results not only for academic researchers but also for industry. Therefore, I recommend it for publication with minor revision. Please find the the comments below:

L34: I suggest to change ‘aqua culturist’ into ‘farmers’ or ‘fish farmers’

L86: The sentence should be slightly modifies ‘…as an ingredient/ substitute to fishmeal..” and ‘aquafeeds’ not ‘aqua feed’

L326-346: place only one ‘(Table 8)’ at the end of this paragraph

I hope the figure 1 will be presented in better resolution in the revised version because it is hard to see the small print.

Conclusions are a bit too long. I advise to make it more concise.

Author Response

Reviewer 1

Comments and Suggestions for Authors

The manuscript entitled: ‘Black Soldier Fly Larvae meal as alternative to fish meal for aquaculture feed’ is well-written, interesting manuscript presenting original and useful results not only for academic researchers but also for industry. Therefore, I recommend it for publication with minor revision. Please find the comments below:

L34: I suggest to change ‘aqua culturist’ into ‘farmers’ or ‘fish farmers’. Changed in fish farmers (line 35)

L86: The sentence should be slightly modifies ‘…as an ingredient/ substitute to fishmeal..” and ‘aquafeeds’ not ‘aqua feed’. Changed in a substitute to fish meal in aquafeeds (line 91)

L326-346: place only one ‘(Table 8)’ at the end of this paragraph. Done (line 370)

I hope the figure 1 will be presented in better resolution in the revised version because it is hard to see the small print. In attachment we are sending the Fig 1 in jpeg and TFF forma

Conclusions are a bit too long. I advise to make it more concise. Done

Reviewer 2 Report

Dear Authors of the manuscript " Black Soldier Fly Larvae meal as alternative to fish meal for aquaculture feed" I carefully read your article. 

This is only a preliminary part of a more large study, where I suppose that you will use the feed, tested here for their chemical composition, in a feeding trial with Seabream. When you will have done this part I will be glade to reconsider my decision. 

Author Response

AUTHORS REPLY

Reviewer 2

Comments and Suggestions for Authors

Dear Authors of the manuscript " Black Soldier Fly Larvae meal as alternative to fish meal for aquaculture feed" I carefully read your article. 

This is only a preliminary part of a more large study, where I suppose that you will use the feed, tested here for their chemical composition, in a feeding trial with Seabream. When you will have done this part I will be glade to reconsider my decision. Yes, this study is a part of a Research project “Feed Insects For Aquaculture” acronym FIFA, PO FEAM 2014/2020 – Mis. 2.47 Innovation in aquaculture, CUP J46C18000570006 Codex 03/INA/17, and I am the Scientific Responsible. The feeding trial was carried out on Sparus aurata.

Reviewer 3 Report

Oteri et al. explored the potential of black soldier fly larvae-based meal as an alternative source of protein in aquaculture feed. This work has merit because it brings a lot of hands-on information and data on the potential substitution of fish-based meal by insect-based, specifically BSF, protein sources and a very thorough characterization of the ingredients has been performed by the authors. The introduction in several points is more broad and generalized, whereas I would expect some better background on the insect/BSF-based alternatives for aquaculture. However, the discussion part is quite strong on literature citations and well documented arguments that support the observed results.

I would strongly suggest that an English native speaker revises the text, there are several minor, yet important, syntax issues scattered through the manuscript. I have posted only a few corrections in the first part but there are many more which need to be inspected.

Please find my more specific points below:

Line 17: please define the ISO abbreviation

Line 24: I would recommend that the authors use one more line to explain ¨good microbiological quality¨

Line 41: Please define where price increase refers. Probably fishmeal I guess.

Lines 43-45: This sentence in fact replicates the previous. Consider removing.

Lines 48-49: Try to rephrase or redefine ¨co-product¨ and ¨ex-food or former food products¨. It is a bit vague to define what these are.

Line 51: ¨rediscovered¨ implies that it once used to be prevalent and then it stopped till recently. Is this the case with insect meals?

Line 54: At this point, it would be useful to cite some relevant previous research papers.

Line 56: ¨can be used¨

Line 57: ¨to be added¨

Lines 57-61: I would say that these sentences do not add substantial info to the background.

Lines 65-68: I would kindly ask the authors to rephrase this part because it is quite difficult to apprehend the message they want to convey.

Line 71: May I ask if there are any commercial fish feed products that contain some % of HIM?

Line 75: ¨pathogen-free feeds to the fish¨

Line 76: So, in which way is extrusion processing insect meals? I would revise the ¨such as fish feeds¨ part.

Line 80: Well maybe ¨devoid of light¨ is a bit of an overstretch particularly for cultured fish species which typically are not found in the aphotic zone.

Line 90: I can imagine that this part refers to the palatability of the HIM-containing fish meals.

A more general question on the introduction; How much the digestibility of such HIM-based fish meals can be affected? Is there any background on this topic?

As mentioned earlier, I would recommend including some more specific background on the insect-based alternatives in aquaculture and make the introduction more focused on the topic. There is a lot of research done in the literature eg. https://www.sciencedirect.com/science/article/pii/S0044848618322208

https://www.ncbi.nlm.nih.gov/pmc/articles/PMC7697048/

https://www.nature.com/articles/s41598-019-45172-5

Lines 171-185: Make sure that the format in this part is the same as in the previous ones.

Line 266: Could please provide some additional information on how this DI is determined?

Line 258-267: Same as format comment above.

Line 279: I would suggest that the authors cite the sources of these indices in Materials and Methods section, just to facilitate the reader’s flow.

Lines 326-346: Same as format comment above.

Line 346: isn’t lead significantly different even for HIM50%?

Line 375: Maybe this is due to the pdf format but the quality of the PCA figure is very poor. I would strongly recommend that the figure is revised in a much higher quality.

Lines 368-370: Is there any particular reason that data from these sensors were chosen?

Line 378: By ¨the first two¨, do you mean e-nose and e-tongue data?

Line 379: I can get the point of their differences on the plot, but wasn’t it expected to get such a discrimination as you change the ingredients? Could you please further elaborate on the biological meaning of these results?

Discussion: Same as format comment above.

Line 400: Were there any difference in the extrusion methods compared to previous research?

Line 402: Please elaborate on the feeding substrate justification.

Line 486: May I ask if there is any disinfection/pasteurization process during feed extrusion process?

Line 533: It seems that insect meal have a strong potential to, partially, substitute fish meal. However, are there any information about digestibility of the HIM-based feed?

Line 553: Are the fish trials with growth rates, performance, health effects etc. published or going to be any time soon? Over formulation and presumptive organic waste could as well be issues of great concern in terms of aquaculture nutrition.

Author Response

AUTHORS REPLY

Reviewer 3

Comments and Suggestions for Authors

Oteri et al. explored the potential of black soldier fly larvae-based meal as an alternative source of protein in aquaculture feed. This work has merit because it brings a lot of hands-on information and data on the potential substitution of fish-based meal by insect-based, specifically BSF, protein sources and a very thorough characterization of the ingredients has been performed by the authors. The introduction in several points is more broad and generalized, whereas I would expect some better background on the insect/BSF-based alternatives for aquaculture. However, the discussion part is quite strong on literature citations and well documented arguments that support the observed results.

I would strongly suggest that an English native speaker revises the text, there are several minor, yet important, syntax issues scattered through the manuscript. I have posted only a few corrections in the first part but there are many more which need to be inspected. Done

Please find my more specific points below:

Line 17: please define the ISO abbreviation. Done: International Organization for Standardization (line 17)

Line 24: I would recommend that the authors use one more line to explain ¨good microbiological quality¨ Done: “Diets showed good hygienic and sanitary quality”.  (The abstract should be a total of about 200 words maximum) (line 25)

Line 41: Please define where price increase refers. Probably fishmeal I guess. Yes, the price is referred to fishmeal. We added in the sentence. (line 43)

Lines 43-45: This sentence in fact replicates the previous. Consider removing. Yes, deleted

Lines 48-49: Try to rephrase or redefine ¨co-product¨ and ¨ex-food or former food products¨. It is a bit vague to define what these are. Here the distinction among ¨co-product¨ and ¨ex-food or former food products¨.

The distinction between by- and co-products is not always consistent and it usually takes place outside the animal feed industry, as it is related to the objectives of other parties involved in food processing [Crawshaw, 2001]. In accordance with Article 5 of the Revised Waste Framework Directive [European Union. Directive No 2008/98], food industry by-products can be defined as materials: resulting from a production process, the primary aim of which is not the production of that item only if the following conditions are met: (a) further use of the substance or object is certain; (b) the substance or object can be used directly without any further processing other than normal industrial practice; (c) the substance or object is produced as an integral part of a production process; and (d) further use is lawful, i.e., the substance or object fulfils all relevant product, environmental and health protection requirements for the specific use and will not lead to overall adverse environmental or human health impacts. Plant materials exposed to various physical and chemical treatments for the extraction of economically important components can be characterized as co-products [Serena et al., 2007]. In this respect, citrus pulp—the solid residue that remains after squeezing the fruit for juice—is a by-product, whereas citrus molasses—the syrup produced by the concentration of juice released from the citrus peel—is a co-product.

  1. Crawshaw, R. Co-Product Feeds: Animal Feeds from the Food and Drinks Industries; Nottingham University Press: Nottingham, UK, 2001.
  2. European Union. Directive No 2008/98 of the European Parliament and the Council of 19 November 2008 on Waste and Repealing Certain Directives. J. Eur. Union 2008, L312, 3–30.
  3. Serena, A.; Knudsen, K.E. Chemical and physicochemical characterisation of co-products from the vegetable food and agro industries. Feed Sci. Technol. 2007, 139, 109–124.

By definition, “Ex-food” or “Former foodstuffs” (FFPs) means foodstuffs which were manufactured for human consumption in full compliance with the EU food law, but which are no longer intended for human consumption for practical or logistical reasons and which do not present any health risks when used as feed. Examples of FFPs include various leftovers from the food industry: pasta, bread, cereals, savoury snacks, biscuits, sweets and chocolate bars. Such foods are rich in sugar, starch, oil or fat, thus giving them a high energy content.

  1. Pinotti, L.; Giromini, C.; Ottoboni, M.; Tretola, M.; Marchis, D. Review: Insects and former foodstu↵s for upgrading food waste biomasses/streams to feed ingredients for farm animals. Animal 2019, 13, 1365–1375.
  2. Giromini, C.; Ottoboni, M.; Tretola, M.; Marchis, D.; Gottardo, D.; Caprarulo, V.; Baldi, A.; Pinotti, L. Nutritional evaluation of former food products (ex-food) intended for pig nutrition. Addit. Contam. Part. A 2017, 34, 1436–1445. [CrossRef] [PubMed]
  3. Tretola, M.; Di Rosa, A.; Tirloni, E.; Ottoboni, M.; Giromini, C.; Leone, F.; Bernardi, C.E.M.; Dell’Orto, V.; Chiofalo, V.; Pinotti, L. Former food products safety: Microbiological quality and computer vision evaluation of packaging remnants contamination. Addit. Contam. Part. A 2017, 34, 1427–1435. [CrossRef]
  4. Pinotti, L.; Ottoboni, M.; Luciano, A.; Savoini, G.; Cattaneo, D.; Tretola, M. Ex-food in animal nutrition: Potentials and challenges. In Energy and Protein Metabolism and Nutrition; EAAP Publication n. 138; Chizzotti, M.L., Ed.; Wageningen Academic Publishers: Wageningen, The Netherlands, 2019; pp. 47–52. [CrossRef]

Line 51: ¨rediscovered¨ implies that it once used to be prevalent and then it stopped till recently. Is this the case with insect meals? Rephrased the sentence  (lines 51-52)

Line 54: At this point, it would be useful to cite some relevant previous research papers. Yes, as I changed the previous sentence, I added the literature suggested by you (references 8, 9)

Line 56: ¨can be used¨ Done (line 58)

Line 57: ¨to be added¨ Done (line 59)

Lines 57-61: I would say that these sentences do not add substantial info to the background. Deleted

Lines 65-68: I would kindly ask the authors to rephrase this part because it is quite difficult to apprehend the message they want to convey. Done, this sentence was rephrased (lines 65-71)

Line 71: May I ask if there are any commercial fish feed products that contain some % of HIM? The companies I contacted, leaders in the production of aquaculture feed, told me that they did not produce feed containing insect meal. The feed in this study is a production that amounts to 160 kg for a small-scale experimental test on sea bream. This feed was prepared by Sparos LDA. SPAROS produces tailor-made feeds for fish and shrimp in its pilot feed mill dedicated to R&D projects. Sparos provides product development services that take advantage of expertise in aquafeed formulation, a wide range of raw materials in stock, and a customised technology that allows production at small scales (batches from 10 to 500 kg).

Line 75: ¨pathogen-free feeds to the fish¨. Changed (line 81)

Line 76: So, in which way is extrusion processing insect meals? I would revise the ¨such as fish feeds¨ part. We rephrased this sentence (lines 81-84)

Line 80: Well maybe ¨devoid of light¨ is a bit of an overstretch particularly for cultured fish species which typically are not found in the aphotic zone. We deleted this sentence (line 85)

Line 90: I can imagine that this part refers to the palatability of the HIM-containing fish meals. Yes, we added “on the HIM-containing fish meals” (line 96)

A more general question on the introduction; How much the digestibility of such HIM-based fish meals can be affected? Is there any background on this topic? I added a sentence on the digestibility of HIM-based fish meal (lines 71-74)

As mentioned earlier, I would recommend including some more specific background on the insect-based alternatives in aquaculture and make the introduction more focused on the topic. There is a lot of research done in the literature eg.

https://www.sciencedirect.com/science/article/pii/S0044848618322208

https://www.ncbi.nlm.nih.gov/pmc/articles/PMC7697048/

https://www.nature.com/articles/s41598-019-45172-5

Yes, I included the following references:

Mateusz Rawski, Jan Mazurkiewicz, Bartosz Kieron ́czyk and Damian Józefiak.  Black Soldier Fly

Full-Fat Larvae Meal as an Alternative to Fish Meal and Fish Oil in Siberian Sturgeon Nutrition: The

Effects on Physical Properties of the Feed, Animal Growth Performance, and Feed Acceptance and

Utilization Animals 2020, 10, 2119; doi:10.3390/ani10112119

Ikram Belghit, Nina S. Liland, Petter Gjesdal, Irene Biancarosa, Elisa Menchetti, Yanxian Li, Rune Waagbø, Åshild Krogdahl, Erik-Jan Lock. Black soldier fly larvae meal can replace fish meal in diets of sea-water phase Atlantic salmon (Salmo salar). Aquaculture 503 (2019) 609-619

Lines 171-185: Make sure that the format in this part is the same as in the previous ones. Yes, done (lines 188-203)

Line 266: Could please provide some additional information on how this DI is determined? Yes, we added some additional information on DI (lines 284-288).

Line 258-267: Same as format comment above. Yes, done (lines 276-289)

Line 279: I would suggest that the authors cite the sources of these indices in Materials and Methods section, just to facilitate the reader’s flow. Yes, sorry, we have added the formulae for AI, TI, PI and H/H (lies 133-141)

Lines 326-346: Same as format comment above. Yes, done (lines 349-370)

Line 346: isn’t lead significantly different even for HIM50%? Yes, sorry, we corrected the significance for lead (lines 369-370 and Table 8)

Line 375: Maybe this is due to the pdf format but the quality of the PCA figure is very poor. I would strongly recommend that the figure is revised in a much higher quality. In attachment we are sending the Fig 1 in jpeg and TFF format

Lines 368-370: Is there any particular reason that data from these sensors were chosen? Yes, we added the sentence “Data from sensors with the highest discrimination power were chosen” (lines 393-394)

Line 378: By ¨the first two¨, do you mean e-nose and e-tongue data? We refer to the first two components of principal component analysis. The first principal component can equivalently be defined as a direction that maximizes the variance of the projected data.

Line 379: I can get the point of their differences on the plot, but wasn’t it expected to get such a discrimination as you change the ingredients? Could you please further elaborate on the biological meaning of these results?

A small percentage substitution of a single ingredient not always lead to sensory changes in the product. In our study, sensors responses showed a discriminant capacity according to the chemical profile of different groups, mostly in relation to the amino acidic profile of the diets. These results are interesting from a biological point of view because in different fish species, feeding behaviour is triggered by different chemical substances which may act as attractants via olfaction or taste. Fish, in general, have a well-defined olfactory sensitivity to amino acids; this phenomenon is generally considered to be involved in the location and identification of food. See the paragraph of Discussion.

Discussion: Same as format comment above. Yes, done

Line 400: Were there any difference in the extrusion methods compared to previous research? Feeds were prepared by SPAROS using an extrusion processing and I do not know exactly the extrusion method applied by SPAROS. Sparos declared:

  • Pellet size: 4 mm
  • Manufacture by extrusion and vacuum coating of oils

Line 402: Please elaborate on the feeding substrate justification. We rephrased the sentence (lines 424-427)

Line 486: May I ask if there is any disinfection/pasteurization process during feed extrusion process? No, there is not. It is well-known that temperature and pressure involved in extrusion cooking inactivates naturally occurring toxins (mycotoxins, glycoalkaloids and allergens) and nutritionally active factors (trypsin inhibitors, gossypol, ...). It also eliminates contaminating micro-organisms.

Line 533: It seems that insect meal have a strong potential to, partially, substitute fish meal. However, are there any information about digestibility of the HIM-based feed? Yes, there are. I added some information in the Introduction paragraph (lines 72-74, reference 14).

Line 553: Are the fish trials with growth rates, performance, health effects etc. published or going to be any time soon? Over formulation and presumptive organic waste could as well be issues of great concern in terms of aquaculture nutrition. Yes, this study is a part of a Research project “Feed Insects For Aquaculture” acronym FIFA, PO FEAM 2014/2020 – Mis. 2.47 Innovation in aquaculture, CUP J46C18000570006 Codex 03/INA/17, and I am the Scientific Responsible. The trial was carried out on Sparus aurata.

Round 2

Reviewer 2 Report

Dear Authors, you confirmed that a feeding trial on seabream was carried out, so I think that it will be more correct to present the data of present article toghether with results of this feeding trial. The palatability of your experimental feed, measured in vitro in manuscript was confirmed by seabream feed intake? What about of sea bream fed by experimental feed muscle composition?

I am sorry but I cannot approve the breaking up of a single trial in many articles which will result all incomplete. This is a so called "salami science" and it is not an adequate way to disseminate the acquired knowledge.

Author Response

Reviewer 2

Dear Authors, you confirmed that a feeding trial on seabream was carried out, so I think that it will be more correct to present the data of present article toghether with results of this feeding trial. The palatability of your experimental feed, measured in vitro in manuscript was confirmed by seabream feed intake? What about of sea bream fed by experimental feed muscle composition?

I am sorry but I cannot approve the breaking up of a single trial in many articles which will result all incomplete. This is a so called "salami science" and it is not an adequate way to disseminate the acquired knowledge.

Dear Reviewer,

I understand your position, but the aim of the manuscript entitled: ‘Black Soldier Fly Larvae meal as alternative to fish meal for aquaculture feed’ is to provide hands-on information and data on the potential substitution of fish-based meal by insect-based, specifically BSF, protein sources through a characterization of the feeds which can be useful not only for academic researchers but also for industry. Knowledge of the chemical and microbiological characteristics of the feed is the first step before the feeding trial, probably the most important step, also to explain the in vivo and post mortem performance of fish.

Here, some preliminary results of the feeding trial on Sparus aurata. I recommend you to treat the data with the utmost confidentiality.

Table– In vivo performance of Sparus aurata (LSM± SEM, P)

DIET

HIM0

HIM25

HIM35

HIM50

SEM

P

IBW

143.64

143.59

143.60

143.78

2.883

1.000

FBW

386.64

385.59

396.00

394.19

5.516

0.437

DIR

18.30

18.43

17.77

17.56

0.277

0.156

FCR

1.42

1.43

1.41

1.42

0.029

0.976

SGR

0.76

0.76

0.76

0.74

0.013

0.723

PER

1.82

1.80

1.77

1.76

0.039

0.715

IBW: Initial Body Weight; FBW: Final Body Weight; DIR: Daily Intake Rate % ; FCR: Feed conversion rate  ; SGR: Specific Growth Rate; PER: Protein Efficiency Ratio.

Table – Chemical composition of the fillets of Sparus aurata

(LSM± SEM, P)

DIET

HI0

HI25

HI35

HI50

SEM

P

Moisture

66.04B

66.80b

67.64A

67.96aA

0.290

< 0.0001

Proteins

20.41

20.00

20.07

19.82

0.181

0.141

Lipids

10.79

11.07

11.51

10.85

0.274

0.251

Ash

1.69

1.65

1.66

1.67

0.049

0.917

A,B significantly different for P<0.01 and a,b for P<0.05

As can be seen in the tables, replacing the protein content of the control diet with Hermetia illucens meal did not compromise the in vivo and post mortem performances of Sparus aurata. We believe some of the possible explanations are related to the compositional characteristics of the feeds.

As soon as we publish the feeding trial on Sparus aurata, it will be my responsibility to send you a copy through the Editor of Sustainability journal.

Reviewer 3 Report

First, I would like to thank the authors for meticulously addressing all my comments. I think that current rebuttal of the manuscript is quite thorough and robust; therefore I would be very happy to endorse publication.

I must state that I strongly believe that since there are also some in vivo trials in S. aurata it would be very useful to have the results, however, the authors stated that they are going to publish them soon since this is a part of an underway research project.

Comments:

Lines 70-72: Since digestibility is an issue in HIM-based feed, wouldn’t that be accompanied by organic waste and overformulation, which will subsequently increase the cost after all? Maybe that is something that needs to be addressed in the discussion.

Regarding the figure, I still did not receive the higher quality files but I am pretty confident that the editor will have them and make sure the uploaded files are of good quality, so that is fine by me.

Author Response

First, I would like to thank the authors for meticulously addressing all my comments. I think that current rebuttal of the manuscript is quite thorough and robust; therefore I would be very happy to endorse publication.

I must state that I strongly believe that since there are also some in vivo trials in S. aurata it would be very useful to have the results, however, the authors stated that they are going to publish them soon since this is a part of an underway research project.

Comments:

Lines 70-72: Since digestibility is an issue in HIM-based feed, wouldn’t that be accompanied by organic waste and overformulation, which will subsequently increase the cost after all? Maybe that is something that needs to be addressed in the discussion.

Dear Reviewer, thank you for your observation.

The answer to your question represents the aim of this manuscript which is to provide chemical and microbiological characteristics of the feeds containing insect meal. This is the first step before the feeding trial, probably the most important step, also to explain the in vivo and post mortem performance of fish. 

Here, some preliminary results of the feeding trial on Sparus aurata (I recommend you to treat the data with the utmost confidentiality).

SEE PDF FILE

As can be seen in the table, replacing the protein content of the control diet with Hermetia illucens meal did not compromise the in vivo performance of Sparus aurata. Some possible explanations could be proposed:

  • -  A high tolerance level of insect meal for Sparus aurata fish species, also due to the life stage of fish used for the trial (average weight 144 g).

  • -  Feeds containing Hermetia illucens meal show (Table 7 of the manuscript) a better profile of the essential aminoacids than that of fish meal, including the highest content of the essential AAs methionine and lysine. The presence of these essential AAs into the diets could also be a reason why no changes in growth were seen in the current trial.

We shall treat this aspect in the study on the feeding trial on Sparus aurata.
Regarding the figure, I still did not receive the higher quality files but I am pretty confident that the

editor will have them and make sure the uploaded files are of good quality, so that is fine by me.

Sorry, I uploaded the figure in jpeg format on the electronic system.

Round 3

Reviewer 2 Report

Dear Author, 

I'm sure that the experimental diet have shown good result in SeaBream farming, but all of my objections concern the opportunity of pubblish present study separate from the one that will show results of feeding trial thta you showed me in your reply. 

It is a question of ethics in research.

Please read this article:

Supak Smolcić V. Salami publication: definitions and examples. Biochem Med (Zagreb). 2013;23(3):237-41. doi: 10.11613/bm.2013.030. PMID: 24266293; PMCID: PMC3900084.